# Berry Consumption and Sleep in the Adult US General Population: Results from the National Health and Nutrition Examination Survey 2005–2018

**DOI:** 10.3390/nu15245115

**Published:** 2023-12-15

**Authors:** Li Zhang, Joshua E. Muscat, Penny M. Kris-Etherton, Vernon M. Chinchilli, Julio Fernandez-Mendoza, Laila Al-Shaar, John P. Richie

**Affiliations:** 1Department of Public Health Sciences, Penn State Cancer Institute, Penn State College of Medicine, Pennsylvania State University, Hershey, PA 17033, USA; jrichie@pennstatehealth.psu.edu; 2Department of Nutritional Sciences, Pennsylvania State University, University Park, PA 16802, USA; pmk3@psu.edu; 3Department of Public Health Sciences, Penn State College of Medicine, Pennsylvania State University, Hershey, PA 17033, USA; vchinchilli@pennstatehealth.psu.edu (V.M.C.);; 4Sleep Research & Treatment Center, Department of Psychiatry & Behavioral Health, Penn State Health Milton S. Hershey Medical Center, Penn State University College of Medicine, Hershey, PA 17033, USA; jfernandezmendoza@pennstatehealth.psu.edu

**Keywords:** berries, sleep, sleep duration, sleep difficulty, diet, antioxidant, NHANES

## Abstract

Introduction: Poor sleep is associated with numerous adverse health outcomes. Berries are rich in micronutrients and antioxidants that may improve sleep quality and duration. We determined the association of berry consumption and sleep duration and sleep difficulty among adult participants in NHANES. Methods: We analyzed the diet of US adults aged ≥ 20 y using two non-consecutive 24 h recalls from the National Health and Nutrition Examination Survey 2005 to 2018 (*N* = 29,217). Poor sleep quality was measured by sleep duration (short sleep duration: <7 h), long sleep (≥9 h), and reported sleep difficulty. The relative risk of poor sleep outcomes for berry consumers vs. nonconsumers was modelled using population weight-adjusted multivariable general logistic regression. Results: About 46% of participants reported inadequate sleep duration, and 27% reported sleep difficulties. Twenty-two percent reported consuming berries. Berry consumers had a 10–17% decreased risk of short sleep. The findings were consistent for specific berry types including strawberries and blueberries (*p* < 0.05). No significant associations with long sleep were found for total berries and any berry types. A decreased risk of sleep difficulties was found to be linked to blackberry consumption (adjusted OR = 0.63, 95% CI: 0.40–0.97; *p* = 0.036) but not for other berries. Conclusions: US adult berry consumers had a decreased risk of reporting short sleep compared to nonconsumers. Berries are underconsumed foods in the US adult population, and increased berry consumption may improve sleep quality.

## 1. Introduction

In the United States, about one in three adults does not sleep for the recommended time of 7–9 h nightly [1,2]. Other reports show that 35–50% of adults suffer transient or chronic insomnia [3]. These common sleep complaints can impair quality of life, cause economic burdens, and contribute to physical and mental decline and many chronic diseases [4,5]. Short sleep duration (<7 h), long sleep duration (9 h or longer), and insomnia are all associated with the increased risk/prevalence of cardiovascular diseases [6,7,8,9]. Short (<6 h) and long sleep (≥9 h) durations have been linked to inflammation, metabolic syndrome, obesity, stroke, diabetes, cancer, and mortality [8,10,11,12,13,14,15,16,17]. Short sleepers are more likely to develop hypertension and less likely to report good overall wellbeing [18,19,20], whereas long sleepers have a greater prevalence of underlying mental health concerns: mood disorders, schizophrenia, and depression [21,22].

There has been interest in berries and sleep quality because berries contain melatonin (a natural sleep hormone), antioxidants, essential nutrients (e.g., potassium, vitamin C, calcium, iron, and selenium), and polysaccharides that have been shown to benefit sleep quality [23,24,25,26,27,28]. There have been sleep quality studies in experimental settings; however, the sample sizes have been relatively small [29,30,31]. Furthermore, population studies are limited. One population-based study examined berry consumption and sleep duration. The UK women’s cohort study (UKWCS) of fruits and vegetables, including berries (*n* = 13,958), used a food frequency questionnaire at baseline and a 4-day food diary during four years of follow-up [32]. Raspberries and strawberries and total polyphenol content of the diet were linearly associated with fewer minutes per day of sleep. These findings were unexpected. Sleep difficulty, a measure of sleep quality, is sometimes termed prolonged sleep latency, a symptom of insomnia [33,34]. The present study was conducted to clarify the relationship of berry consumption with sleep quality using categories of high- and low-risk groups of sleep duration and sleep difficulty.

The National Health and Nutrition Examination Survey (NHANES) provides an opportunity to assess the association of berry intake with sleep risk measures. We used 14 years of data, from 2005–2018, to examine the association between berry (and berry subtype) consumption and the risk of short sleep, long sleep, and sleep difficulty among US non-pregnant, non-lactating adults (age ≥ 20 years). We hypothesized that berry consumption may be inversely associated with short sleep, long sleep, and sleep difficulty (i.e., prolonged sleep latency), respectively.

## 2. Methods

### 2.1. Study Design

Initiated in the 1960s, the NHANES, a major program of the National Center for Health Statistics (NCHS), a part of the Centers for Disease Control and Prevention, was designed to assess Americans’ health and nutritional status based on a combination of interviews and physical examinations [35]. Every year starting in 1999, the NHANES recruits about 5000 individuals located in counties across the 50 states to represent the noninstitutionalized US population of all ages. The recruitment strategy was created with a multi-stage probability sampling design covering family, county, state, and region stages. The interviews and physical examinations collected the participants’ demographic, lifestyle, health, and dietary intake information. The NCHS research review board approved NHANES, and data were published in a 2-year cycle. All the participants provided written informed consent. This study was exempt from the Penn State institutional review board approval for using secondary analysis from publicly available data sources.

### 2.2. Analytic Sample

The analyses were conducted based on combining seven cycles of the NHANES data (2005–2006 to 2017–2018). The analytical sample included adults (20 years or older), excluding pregnant women (*n* = 554), lactating women (*n* = 244), and participants who reported missing data on sleep duration or sleep difficulty (*n* = 144), or who had extreme daily energy intake (*n* = 1286; <800 or >4200 kcal for males and <500 or >3500 kcal for females). Trained staff collected up to two 24 h food recalls from the participants with the Automated Multiple-Pass Method proposed by the US Department of Agriculture to enhance accuracy [36,37]. The trained interviewers collected the first recall in person, and 3–10 days later, the second food recall was administered by phone. The analyzed sample was restricted to respondents with two 24 h recalls (*N* = 29,217) (Appendix A).

### 2.3. Definition of Berry Consumption and Consumers

Berries can be consumed individually and are often a component in mixed or processed foods. For this reason, we developed an algorithm to identify berry intake from the food records. For food items that contained berries as part of a food group (e.g., fruit salad), it was necessary to manually search the Food and Nutrient Database for Dietary Studies (FNDDS) food code description further [38,39]. There are two food code descriptions. We identified berries and berry subtypes such as strawberries, blueberries, cranberries, raspberries, and blackberries as ingredients in food item/recipe using main food code descriptions and the additional FNDDS food code description that provides more additional details on each item. Each of these were from the cycle-specific releases of the FNDDS, which are based on corresponding NHANES cycles; details can be found in previously published methods [40,41]. Berry-flavored alcoholic beverages, typically nutritionally distinct from other berry-containing foods, were excluded.

The amount of berries (berry subtypes) (in grams) in each food recipe/item as identified from the FNDDS was then quantified in cup equivalents using cycle-specific releases of the USDA’s Food Patterns Equivalent Database (FPED) [42]. This database converts all reported foods and beverages in dietary components from NHANES into cup-equivalent units for assessing alignment with the USDA 2015–2020 dietary guidelines for Americans, which are based on cup equivalents [43]. In FPED, there is a category called “citrus, melons, and berries” which was used for conversion.

Berry consumers were identified and classified as respondents who reported >0 cup-equivalent intakes of berry (or berry subtype) fruits in either one or both food recalls.

### 2.4. Short/Long Sleep Duration and Sleep Difficulty

Based on the answers to the survey question “How much sleep do you usually get at night on weekdays or workdays”? from the sleep disorders questionnaires of NHANES, sleep duration was categorized into three durations: <7 h as short sleep, 7–9 h as sufficient/adequate sleep, and ≥9 h as long sleep. These categories were based on recommended hours of sleep for the adult population by the American Academy of Sleep Medicine and Sleep Research Society and aligned with prior research indicating an association of insufficient sleep (e.g., <7 h) and long sleep (e.g., >9 h) with chronic diseases [44,45]. Sleep difficulty, a symptom of insomnia, was treated as a binary variable and derived from the answers to the question, “Have you ever told a doctor or other health professional that you have trouble sleeping”?

### 2.5. Covariates

Covariates were selected based on the literature on factors that affect sleep duration and sleep difficulty. A five-level race/ethnicity variable (non-Hispanic White, non-Hispanic Black, Mexican American, other Hispanic, and other), a four-level educational attainment variable (less than high school, high school, some college, 4-year college or more), and a three-level poverty-to-income ratio variable (<1.3, 1.3–1.85, and >1.85) were created to represent the general racial/ethnic spectrum and educational and financial status of the US adult population. Physical activity was treated as a categorical variable with four levels: sedentary (no reported work or leisure physical activity), low (below minimum recommendations), moderate (150–300 min of moderate-intensity or 75–150 min of vigorous activity as recommended), or high (exceeding the level recommended in moderate level). Depression severity in three levels was defined based on PHQ-9 scores: mild (0–3), moderate (4–14), and severe (15–27). The binary variables included sex (Female/Male), and current smoking status (Yes/No). Three-level modified HEI-2015 (a summary diet quality index aligned with dietary guidelines without the berry components) was categorized by quartiles: inadequate: <Q1(43.8), average: Q1–Q3, and optimal: >Q3 (62.5). Body mass index (BMI) was defined as a quotient of weight (kg) and height squared (m^2^). Age (years), BMI (kg/m^2^), total energy intake (kcal/day), alcohol consumption (g/day), and caffeine consumption (mg/day, in logarithmic form) were considered continuous variables.

### 2.6. Statistical Analysis

All analyses were performed with SAS (version 9.4) and reported at a 2-tailed α-level of 0.05 [46]. Appropriate survey sample weights, strata, and primary sampling units were accounted for using survey procedures suggested by NCHS [47]. The comparison of consumers and nonconsumers was based on participants who had provided two dietary recalls, and therefore, the day 2 dietary weights were adjusted for complex study design and non-response, as suggested by NHANES analytical guidelines [48]. A single imputation was performed with the hot-deck technique for missing values of the demographic and lifestyle variables using PROC SURVEYIMPUTE. The Rao-Scott X^2^ test (a design-modified version of Pearson X^2^ test) was adopted to compare the categorical demographic and lifestyle characteristics between consumers and nonconsumers. The weight and study-design adjusted t-test was performed to compare continuous variables.

Multivariable logistic regression analysis using the SAS procedure PROC SURVEYLOGISTIC was used to estimate the association between total berry and individual berry consumption and sleep complaints, namely sleep difficulty, short sleep, or long sleep. The models were used to obtain prevalence odds ratios (ORs) and their 95% confidence intervals (CIs).

A backward stepwise selection procedure was implemented in the model to include categorical variables: sex, race/ethnicity, education level, poverty-to-income ratio, physical activity, current smoking status, depression severity, and modified HEI-2015, as well as continuous variables: age (years), BMI, alcohol consumption, total energy, and caffeine intake. All the covariates were selected based on the prior research [32,49,50,51], as well as the selection criteria: a 10% or more in percentage change between measures of association before and after adjusting for a potential confounder in the model [52].

## 3. Results

### 3.1. Sample Characteristics by Berry Consumption Status

Study subject characteristics are reported in Table 1. Of the eligible respondents (*N* = 29,217), approximately 22% of participants (*n* = 6417) consumed berries (>0 cup equivalents). A higher proportion of berry consumers was women, non-Hispanic White, wealthier, and had a higher level of physical activity, diet quality, and educational attainment. A lower percentage of berry consumers had a history of diabetes or suffered severe depression. Berry consumers also had higher total energy intake, lower intakes of alcohol and caffeine, lower mean BMI, and were less likely to be current smokers than nonconsumers.

### 3.2. Sample Characteristics by Short Sleep, Long Sleep, and Sleep Difficulty

Table 2 and Table 3 summarize the distribution of short sleep, long sleep, and sleep difficulty according to subject characteristics. Compared to adequate sleepers (54.4%), a lower proportion of short sleepers (21.2%) and long sleepers (24.4%) consumed berries (Table 2). There was no difference in proportion of berry consumers by sleep difficulty status (25.5% vs. 25.4%, Table 3). The distribution of short sleep, long sleep, and sleep difficulty differed by sociodemographic, lifestyle, and dietary factors. A higher proportion of the respondents who reported short sleep or long sleep durations or sleep difficulty earned less income, had less education, reported higher BMI, smoked, and more severe depression symptoms. Compared to the respondents with adequate sleep durations, respondents with short or long sleep durations were more likely to be Non-Hispanic Black and Hispanic/Latino (Mexican American and other Hispanics), and have lower mean alcohol intake, mean energy intake, and lower diet quality. Short sleepers tended to be younger and have higher mean caffeine intake while long sleepers were more likely to be women, older, exercise less, and have lower mean caffeine intake. Similarly, respondents with sleep difficulty were more likely to be non-Hispanic White, older, less physically active, had lower mean intakes of alcohol and energy and higher mean caffeine intake. However, there was no difference in diet quality between respondents with sleep difficulty and those without.

### 3.3. Berry Consumption Associations with Short but Not with Long Sleep Duration

The models presented in Table 4 compared the odds of short or long sleep duration vs. adequate duration between berry consumers and nonconsumers before and after progressively adjusting for potential confounders. Berry consumers were less likely to report short sleep, an association that was attenuated but remained significant as we controlled for confounders (adjusted OR ranged from 0.75 in the first model to 0.9 in fully adjusted model 3). Consumers of strawberries (adjusted OR ranged from 0.75 to 0.9) and blueberries (OR ranged from 0.69 to 0.83) also had significantly decreased odds of reporting short sleep than nonconsumers. Compared to nonconsumers, consumption of total berries (adjusted OR = 0.8) and blackberries (adjusted OR = 0.44) was associated with lower odds of long sleep when adjusting for sex, race/ethnicity, and age in model 1; however, no significant associations were found for total berries or any berry subtypes in the fully adjusted model 3.

### 3.4. Berry Consumption Associated with Sleep Difficulty

Table 5 reports the results of the models that compared the odds of sleep difficulty between consumers and nonconsumers before and after adjusting for potential confounders. Compared with nonconsumers, berry consumers (adjusted OR = 0.88, 95% CI: 0.80–0.95, *p* = 0.003) and those who consumed blueberries, raspberries, or blackberries had significantly lower odds of reporting sleep difficulty compared to nonconsumers, after adjusting for age, sex, and race/ethnicity in model 1. After further adjusting for sociodemographic, lifestyle, and dietary confounders, only blackberry consumption remained significantly associated with a 37% reduced odds of sleep difficulty (*p* = 0.036).

We did not observe any evidence showing that sex or BMI modified the association between berry consumption and sleep difficulty or short/long sleep, and the addition of melatonin use (*n* = 369 (1.3%)) as a covariate did not change any of the berry relationships with sleep outcomes.

## 4. Discussion

In this nationally representative sample of US adults, we observed a disparity in sociodemographic, lifestyle, and dietary factors between berry consumers and nonconsumers, suggesting that berry consumers were more educated, affluent, and health-conscious compared to nonconsumers. Respondents who reported sleep complaints (short sleep, long sleep, or sleep difficulties), compared to the ones without, were more likely to be economically disadvantaged and less educated. Those who reported sleep complaints also tended to be sedentary, smokers, have greater BMI, and more severe depression, exhibiting poorer lifestyle and health awareness. While short or long sleepers had lower diet quality as compared to sleepers with recommended sleep duration, diet quality did not differ by sleep difficulty status. Further, for the first time, we found that consumers of total berries (especially strawberries and blueberries) had significantly decreased odds of reporting short sleep versus nonconsumers, indicating that berry consumption was associated with adequate sleep duration. Additionally, the odds for blackberry consumers to report sleep difficulty decreased by 36% compared to nonconsumers.

Berry consumption was also associated with a healthier lifestyle, a higher socioeconomic status, and a better overall health in Finnish men [53]. However, this study differed from ours because it did not find an association between education attainment and berry consumption. The better diet quality enjoyed by berry consumers as compared to nonconsumers was a new observation, confirming and expanding the previously reported association between higher self-rating for healthfulness of diet and higher berry consumption [41].

Our data also revealed sociodemographic and lifestyle disparities by sleep duration and sleep difficulty, indicating that respondents with recommended sleep duration and without sleep difficulty tend to be economically, physically, and mentally healthier. Our findings were consistent with the local- and national-level cross-sectional studies with adult populations, showing that lower income and education attainment that were highly correlated with lower socioeconomic status were associated with more sleep complaints and less efficient and adequate sleep [54,55,56]. This could be attributable to work conditions, employee-related factors (long work hours, work schedule, and shift work), and waking activities (long TV watching hours) [57,58,59]. We also found that people with sleep complaints, compared with the ones without, tend to have less physical activity, adverse health outcomes, and more severe depression, suggesting that regular physical activity and better physical and mental health could be beneficial to sleep improvement [60,61,62,63].

We also showed that short or long sleepers were more likely to be non-Hispanic Black and Hispanic/Latino while the Non-Hispanic White group were more likely to experience sleep difficulty, confirming the persistent increased prevalence of short sleep durations experienced by non-Hispanic Black and Hispanic/Latino people from a large cross-sectional study of US adults based on the National Health Interview Survey (NHIS) data (2004–2018) [64]. The racial disparity in short sleep could be attributable to night shift and decreased work schedule flexibility that likely prevent non-Hispanic Black and Hispanic/Latino individuals from sleeping enough [65,66]. Depending on the varying definitions of long sleep, the results regarding racial disparity for long sleep were mixed. This prior NHIS study along with the other population-based cross-sectional study found that long sleep (defined as >9 h of sleep duration) was only more prevalent among non-Hispanic Blacks, not Hispanics in the US [23]. The inconsistent results could be due to the inclusion of 9 h sleep duration for long sleep in our study. While various factors could contribute to the observed racial disparity in long sleep, persistently higher multimorbidity prevalence and unemployment rate could help explain the higher prevalence of long sleepers among non-Hispanic Blacks [67,68]. Furthermore, our finding of racial disparity in self-reported sleep difficulty could be due to geographical and environmental influences and reporting bias [69,70].

Another interesting observation in our study was that diet quality differed by sleep duration but not by sleep difficulty status, supporting the findings from two cross-sectional studies (2011–2016) based on the Swedish EpiHealth cohort and the NHANES in which sleepers with inadequate sleep duration were less likely to follow a healthy diet [71,72].

Significantly lower odds of reporting short sleep by consumers of total berries (especially strawberries and blueberries) versus nonconsumers was a novel finding. As a rich source of nutrients and bioactive compounds, berries (especially strawberries and blueberries) may improve sleep quality and sleep duration. Berries are a source of sleep-promoting nutrients such as dietary fiber, folate, potassium, vitamin C, calcium, iron, selenium, and melatonin [23,24,26,73,74,75]. In addition, berries (especially strawberries and blueberries) contain abundant polyphenols (including flavonoids which are specifically high in strawberries and blueberries) with potent antioxidant properties that have been associated with reduced oxidative stress in the CNS, decreased inflammation, as well as improved endothelial function and blood pressure control [76,77,78,79,80,81,82,83,84], all of which are associated with improved sleep regulation, sleep quality, and/or sleep duration [30,31,85,86,87].

Interestingly, an inverse association of sleep difficulty was only observed for blackberry consumption but not other berries. This could be a chance finding. Blackberries contain relatively high levels of vitamin C, magnesium, and iron, which promote quality sleep [24,88,89,90]. However, blackberries are consumed infrequently compared to other berry types. It may be that blackberry consumers differ in a number of health behaviors that may reduce their risk of sleep difficulty.

In interpreting the results of the present study, there are limitations. Using two 24 h self-reported food recalls for determining berry consumer status could misclassify infrequent berry consumers as nonconsumers. This misclassification, if assumed to be nondifferential, may underestimate the associations between berry consumption and inadequate sleep. Secondly, the temporal associations cannot be determined because the study was cross-sectional. Thirdly, NHANES does not contain sleep-related information including shift work and work conditions, which may account for the associations found in this study. Additionally, the sleep duration measurement only included weekday/workday sleep, which could differentiate the study results if the sleep duration was based on weekday and weekend sleep. In addition, sleep data were self-reported, which can be subjective, non-specific, and reflective of underlying sleep disorders or other health concerns. However, substantial evidence supports that subjective and retrospective reporting of sleep measurements, similar to the ones obtained from objective methods, was arguably reliable for capturing sleep habits and reflective of important associations to health outcomes [91,92,93]. Furthermore, self-reported sleep duration was not correlated with sleep difficulty, which is not uncommon for subjective sleep measurements. Thus, we did not consider the association between berry consumption and combination of self-reported sleep duration and sleep difficulty. Finally, we cannot rule out the possibility that residual confounding can explain the associations found in our observational study, although we adjusted for the crucial confounders (demographic, lifestyle, and dietary factors) to mitigate the potential confounding effects. While our results are encouraging, the exact quantity of berries that may benefit sleep and the mechanisms by which berry consumption may affect sleep need further study in a future clinical trial.

## 5. Conclusions

Berry consumption was associated with decreased odds of short sleep, indicating that berries can, directly or indirectly, promote quality sleep. Because adequate sleep is crucial for the quality of life and disease prevention, berry consumption may benefit health.

## Figures and Tables

**Table 1 nutrients-15-05115-t001:** Weighted characteristics of the NHANES respondents by berry consumption status (2005–2018), *N* = 29,217.

Characteristics	Nonconsumers		Berry Consumers		*p*-Value
	*n* = 22,800		*n* = 6417		
Age, mean ± S.E., y		46.9 ± 0.3		51.1 ± 0.4	<0.0001
BMI, mean ± S.E., kg/$m^{2}$		25.6 ± 0.2		24.0 ± 0.3	<0.0001
Sex (female), %	11,331	49.2 (48.2, 50.2)	3867	60.9 (59.2, 62.6)	<0.0001
Race/ethnicity, %					<0.0001
Non-Hispanic White	9643	64.7 (62.0, 67.5)	3438	76.6 (74.2, 79.1)	
Non-Hispanic Black	5330	12.7 (11.1, 14.3)	988	6.6 (5.6, 7.7)	
Mexican American	3590	9.3 (7.9, 10.8)	759	5.6 (4.6, 6.7)	
Other Hispanic	2186	5.6 (4.6, 6.5)	546	4.3 (3.5, 5.2)	
Other	2231	7.7 (6.9, 8.5)	686	6.8 (5.7, 7.9)	
Poverty-to-income ratio, %					<0.0001
<1.3	7493	24.3 (22.8, 25.7)	1349	13.6 (12.3, 15.0)	
1.3–1.85	8817	36.4 (35.1, 37.7)	2298	31.6 (29.5, 33.7)	
>1.85	6490	39.3 (37.4, 41.3)	2770	54.8 (52.2, 57.3)	
Education, %					<0.0001
Less than high school	5864	17.0 (15.8, 18.2)	881	8.6 (7.4, 9.8)	
High school	5564	25.6 (24.4, 26.8)	1187	16.3 (14.9, 17.7)	
Some college	6785	32.1 (31.0, 33.2)	1936	30.0 (27.8, 32.2)	
≥4-year degree	4587	25.3 (23.5, 27.0)	2413	45.1 (42.4, 47.8)	
Physical activity, %					<0.0001
Sedentary	6097	21.8 (20.7, 22.8)	1304	16.1 (14.6, 17.7)	
Low	3751	16.6 (15.8, 17.4)	1070	15.4 (14.3, 16.5)	
Moderate	3190	14.5 (13.7, 15.3)	1091	18.1 (16.5, 19.7)	
High	9762	47.1 (45.8, 48.4)	2952	50.4 (48.4, 52.4)	
Current smoker (Yes), %	5037	22.7 (21.6, 23.7)	642	9.4 (8.2, 10.6)	<0.0001
Depression severity, %					<0.0001
0–3	15,503	68.9 (67.9, 69.9)	4757	75.0 (73.4, 76.6)	
4–14	6438	27.8 (26.8, 28.7)	1537	23.4 (21.9, 25.0)	
15–27	859	3.3 (3.0, 3.7)	123	1.6 (1.2, 2.0)	
Modified HEI-2015, %					<0.0001
<43.8	6135	29.1 (27.9, 30.3)	818	13.0 (11.6, 14.5)	
43.8–62.5	11,711	50.8 (49.7, 51.9)	2982	47.8 (45.7, 49.8)	
>62.5	4954	20.1 (19.1, 21.2)	2617	39.2 (37.0, 41.5)	
Alcohol intake, g/day		9.0 ± 0.3		8.3 ± 0.4	<0.0001
Energy intake, kcal/day		2031.9 ± 8.3		2057.2 ± 14.3	<0.0001
Caffeine intake, mg/day		167.3 ± 2.9		163.3 ± 3.3	<0.0001

All percentages and means ± S.E. were adjusted for survey weights. Comparison between berry consumers and nonconsumers: for categorical data, *p*-value was estimated using Rao-Scott x^2^ test for categorical data and *t*-test for continuous data.

**Table 2 nutrients-15-05115-t002:** Weighted characteristics of the NHANES respondents by sleep duration (2005–2018), *N* = 29,217.

Characteristics	Short Sleep Duration		Long Sleep Duration		Recommended/Adequate Sleep Duration		*p*-Value
	*n* = 10,159		*n* = 3403		*n* = 15,655		
Age, mean ± S.E., y		47.2 ± 0.3		49.8 ± 0.6		48.1 ± 0.3	<0.0001
BMI, mean ± S.E., kg/$m^{2}$		29.8 ± 0.1		29.0 ± 0.2		28.7 ± 0.1	<0.0001
Sex (female), %	5108	48.8 (47.3, 50.3)	1935	60.5 (57.9, 63.2)	8155	52.3 (51.3, 53.4)	<0.0001
Race/ethnicity, %							<0.0001
Non-Hispanic White	3867	61.3 (58.2, 64.4)	1594	67.0 (63.4, 70.6)	7440	71.5 (69.0, 73.9)	
Non-Hispanic Black	2970	16.8 (14.7, 19.0)	663	10.9 (9.0, 12.9)	2685	8.1 (7.0, 9.2)	
Mexican American	1380	8.4 (7.1, 9.7)	516	9.5 (7.4, 11.6)	2453	8.1 (6.8, 9.4)	
Other Hispanic	1023	6.1 (5.0, 7.2)	315	5.3 (4.0, 6.6)	1394	4.8 (4.0, 5.6)	
Other	919	7.4 (6.5, 8.2)	315	7.3 (5.6, 8.9)	1683	7.5 (6.6, 8.5)	
Poverty-to-income ratio, %							<0.0001
<1.3	3280	24.4 (22.6, 26.1)	1255	28.6 (25.9, 31.2)	4310	18.6 (17.4, 19.8)	
1.3–1.85	3889	36.6 (35.1, 38.1)	1347	37.8 (35.1, 40.6)	5905	34.4 (32.8, 36.1)	
>1.85	2990	39.0 (36.6, 41.3)	801	33.6 (30.1, 37.1)	5440	46.9 (44.9, 49.0)	
Education, %							<0.0001
Less than high school	2421	16.2 (14.9, 17.6)	935	18.9 (17.0, 20.9)	3387	13.3 (12.0, 14.6)	
High school	2513	26.6 (25.0, 28.2)	849	26.4 (24.2, 28.6)	3389	20.7 (19.5, 21.9)	
Some college	3223	34.5 (32.8, 36.1)	996	31.0 (28.3, 33.7)	4505	30.1 (28.7, 31.5)	
≥4-year degree	2002	22.7 (20.9, 24.5)	623	23.7 (20.4, 26.9)	4374	35.9 (33.7, 38.0)	
Physical activity, %							<0.0001
Sedentary	2564	21.0 (19.6, 22.4)	1148	28.7 (26.2, 31.3)	3689	18.2 (17.1, 19.3)	
Low	1656	15.5 (14.6, 16.5)	559	15.7 (13.9, 17.5)	2606	16.9 (15.8, 17.9)	
Moderate	1349	13.8 (12.8, 14.8)	452	13.8 (11.8, 15.7)	2480	16.7 (15.7, 17.6)	
High	4590	49.7 (47.8, 51.5)	1244	41.8 (38.4, 45.2)	6880	48.2 (46.7, 49.7)	
Current smoker (Yes), %	2425	25.3 (23.8, 26.7)	657	20.3 (18.2, 22.4)	2595	15.8 (14.7, 16.9)	<0.0001
Depression severity, %							
0–3	6312	63.1 (61.5, 64.8)	2208	63.8 (61.4, 66.1)	11,779	76.2 (74.5, 77.5)	
4–14	3336	32.4 (31.0, 33.9)	1046	31.7 (29.2, 34.2)	3551	22.1 (20.8, 23.3)	
15–27	511	4.5 (3.8, 5.1)	149	4.5 (3.4, 5.6)	325	1.7 (1.4, 2.0)	
Modified HEI-2015							<0.0001
<43.8	2699	28.0 (26.6, 29.4)	879	28.3 (25.2, 31.4)	3375	22.7 (21.3, 24.0)	
43.8–62.5	5137	50.2 (48.8, 51.6)	1686	49.6 (47.0, 52.3)	7870	49.9 (48.7, 51.2)	
>62.5	2323	21.8 (20.3, 23.3)	838	22.1 (19.5, 24.7)	4410	27.4 (26.0, 28.7)	
Alcohol consumption, g/day		8.3 ± 0.4		8.7 ± 0.6		9.2 ± 0.3	<0.0001
Energy intake, kcal/day		2044.7 ± 11.9		1927.7 ± 18.5		2057.3 ± 9.6	<0.0001
Caffeine intake, mg/day		176.8 ± 4.0		140.4 ± 4.2		165.7 ± 2.9	<0.0001
Berry consumers (Yes), %	1962	21.2 (19.7, 22.6)	719	24.4 (21.8, 27.0)	3736	28.0 (26.4, 29.6)	<0.0001

All percentages and means ± S.E. were adjusted for survey weights. Comparison between berry consumers and nonconsumers: for categorical data, false-discovery rate-adjusted *p*-value was estimated using Rao-Scott x^2^ test for categorical data and *t*-test for continuous data.

**Table 3 nutrients-15-05115-t003:** Weighted characteristics of the NHANES respondents by sleep difficulty (2005–2018), *N* = 29,217.

Characteristics	Sleep Difficulty (Yes)		Sleep Difficulty (No)		*p*-Value
	*n* = 7694		*n* = 21,523		
Age, mean ± S.E., y		51.4 ± 0.3		46.6 ± 0.3	<0.0001
BMI, mean ± S.E., kg/$m^{2}$		30.2 ± 0.1		28.7 ± 0.1	<0.0001
Sex (female), %	4561	59.0 (57.3, 60.8)	10,637	49.6 (48.6, 50.5)	<0.0001
Race/ethnicity, %					<0.0001
Non-Hispanic White	4051	74.5 (72.0, 77.0)	8850	65.2 (62.6, 67.8)	
Non-Hispanic Black	1639	10.3 (8.9, 11.7)	4679	11.5 (10.1, 12.9)	
Mexican American	779	5.2 (4.2, 6.1)	3570	9.6 (8.1, 11.0)	
Other Hispanic	642	4.1 (3.3, 4.9)	2090	5.7 (4.8, 6.7)	
Other	583	5.9 (5.1, 6.8)	2334	8.0 (7.1, 8.9)	
Poverty-to-income ratio, %					0.0196
<1.3	2537	23.3 (21.5, 25.2)	6308	20.9 (19.7, 22.1)	
1.3–1.85	2817	34.3 (32.2, 36.3)	8324	36.0 (34.7, 37.4)	
>1.85	2340	42.4 (39.8, 45.0)	6891	43.1 (41.2, 45.0)	
Education, %					<0.0001
Less than high school	1636	13.3 (11.9, 14.7)	5107	15.5 (14.2, 16.7)	
High school	1833	24.8 (23.1, 26.6)	4918	22.6 (21.5, 23.7)	
Some college	2535	33.9 (32.0, 35.9)	6189	30.7 (29.5, 31.8)	
≥4-year degree	1690	27.9 (25.5, 30.4)	5309	31.3 (29.4, 33.1)	
Physical activity, %					<0.0001
Sedentary	6097	21.8 (20.7, 22.8)	5211	19.1 (18.1, 20.1)	
Low	3751	16.6 (15.8, 17.4)	3495	15.9 (15.2, 16.7)	
Moderate	3190	14.5 (13.7, 15.3)	3211	15.7 (14.8, 16.5)	
High	9762	47.1 (45.8, 48.4)	9606	49.3 (48.0, 50.6)	
Current smoker (Yes), %	1855	23.7 (22.0, 25.4)	3822	17.6 (16.7, 18.5)	<0.0001
Depression severity, %					<0.0001
0–3	3729	51.1 (49.2, 53.0)	16,570	78.1 (77.1, 79.2)	
4–14	3331	42.0 (40.2, 43.8)	4602	20.5 (19.5, 21.5)	
15–27	634	6.9 (6.2, 7.6)	351	1.4 (1.1, 1.6)	
Modified HEI-2015					0.518
<43.8	1981	25.6 (23.8, 27.3)	4972	24.8 (23.5, 26.0)	
43.8–62.5	3820	50.1 (48.4, 51.8)	10,873	49.9 (48.8, 51.1)	
>62.5	1893	24.3 (22.6, 26.1)	5678	25.3 (24.0, 26.5)	
Alcohol intake, g/day		8.7 ± 0.4		8.9 ± 0.2	<0.0001
Energy intake, kcal/day		1976.6 ± 12.6		2061.9 ± 8.2	<0.0001
Caffeine intake, mg/day		180.5 ± 3.3		160.8 ± 2.8	<0.0001
Berry consumers (Yes), %	1684	25.5 (23.7, 27.2)	4733	25.4 (24.1, 26.8)	0.941

All percentages and means ± S.E. were adjusted for survey weights. Comparison between berry consumers and nonconsumers: for categorical data, false-discovery rate-adjusted *p*-value was estimated using Rao-Scott x^2^ test for categorical data and *t*-test for continuous data.

**Table 4 nutrients-15-05115-t004:** Adjusted prevalence odds ratio (95% CI) of berry consumption associated with self-reported long or short sleep duration, *N* = 29,217.

Types of Berries	Case # (Consumers/ Nonconsumers)	Self-Reported Sleep Duration	Adjusted OR (95% CI) for Model 1	*p*-Value	Adjusted OR (95% CI) for Model 2	*p*-Value	Adjusted OR (95% CI) for Model 3	*p*-Value
Berries								
	1962/8197	Short sleep	0.75 (0.68, 0.83)	<0.0001	0.89 (0.80, 0.98)	0.019	0.90 (0.81, 0.993)	0.037
	719/2684	Long sleep	0.80 (0.69, 0.93)	0.004	0.96 (0.83, 1.12)	0.639	1.00 (0.86, 1.17)	0.97
	3736/11,919	Normal sleep	1.00		1.00		1.00	
Strawberries								
	1214/8945	Short sleep	0.75 (0.68, 0.83)	<0.0001	0.88 (0.80, 0.98)	0.014	0.90 (0.81, 0.99)	0.024
	461/2942	Long sleep	0.87 (0.75, 1.02)	0.092	1.04 (0.89, 1.22)	0.627	1.08 (0.92, 1.26)	0.351
	2326/13,329	Normal sleep	1.00		1.00		1.00	
Blueberries								
	703/9456	Short sleep	0.69 (0.60, 0.81)	<0.0001	0.82 (0.70, 0.96)	0.012	0.83 (0.71, 0.97)	0.018
	296/3107	Long sleep	0.82 (0.66, 1.01)	0.061	0.99 (0.80, 1.24)	0.952	1.03 (0.83, 1.28)	0.775
	1494/14,161	Normal sleep	1.00		1.00		1.00	
Cranberries								
	272/9887	Short sleep	0.70 (0.54, 0.91)	0.007	0.84 (0.65, 1.09)	0.182	0.85 (0.66, 1.10)	0.217
	102/3301	Long sleep	0.88 (0.66, 1.19)	0.416	1.09 (0.80, 1.48)	0.595	1.13 (0.83, 1.55)	0.429
	560/15,095	Normal sleep	1.00		1.00		1.00	
Raspberries								
	105/10,054	Short sleep	0.69 (0.45, 1.04)	0.075	0.83 (0.54, 1.27)	0.392	0.84 (0.55, 1.28)	0.416
	35/3368	Long sleep	0.79 (0.48, 1.30)	0.346	0.98 (0.59, 1.63)	0.923	1.00 (0.60, 1.66)	0.995
	222/15,433	Normal sleep	1.00		1.00		1.00	
Blackberries								
	58/10,101	Short sleep	0.58 (0.34, 0.99)	0.045	0.73 (0.43, 1.26)	0.265	0.74 (0.43, 1.29)	0.277
	19/3384	Long sleep	0.44 (0.21, 0.95)	0.034	0.57 (0.26, 1.26)	0.166	0.58 (0.27, 1.29)	0.182
	148/15,507	Normal sleep	1.00		1.00		1.00	

Model 1: adjusted for age, sex, and race/ethnicity. Model 2: further adjusted for education level, poverty-to-income ratio, physical activity, current smoking status, depression severity, BMI, alcohol consumption, total energy, and caffeine intake. Model 3: further adjusted for education level, poverty-to-income ratio, physical activity, current smoking status, depression severity, BMI, alcohol consumption, total energy, caffeine intake, and modified HEI-2015.

**Table 5 nutrients-15-05115-t005:** Adjusted prevalence odds ratio (95% CI) of berry consumption associated with self-reported sleep difficulty, *N* = 29,217.

Types of Berries	Case # (Consumers/Nonconsumers)	Sleep Difficulty	Adjusted OR (95% CI) for Model 1	*p*-Value	Adjusted OR (95% CI) for Model 2	*p*-Value	Adjusted OR (95% CI) for Model 3	*p*-Value
Berries	1684/6010	Yes	0.88 (0.80, 0.95)	0.003	0.989 (0.90, 1.09)	0.814	0.992 (0.90, 1.09)	0.864
	4733/16,790	No	1.00		1.00		1.00	
Strawberries	1022/6672	Yes	0.92 (0.83, 1.03)	0.128	1.04 (0.93, 1.17)	0.501	1.05 (0.93, 1.18)	0.436
	2979/18,544	No	1.00		1.00		1.00	
Blueberries	662/7032	Yes	0.88 (0.77, 0.99)	0.045	0.96 (0.83, 1.10)	0.535	0.98 (0.85, 1.13)	0.825
	1831/19,692	No	1.00		1.00		1.00	
Cranberries	274/7420	Yes	0.92 (0.73, 1.16)	0.476	1.05 (0.83, 1.33)	0.683	1.07 (0.85, 1.36)	0.564
	660/20,863	No	1.00		1.00		1.00	
Raspberries	96/7598	Yes	0.66 (0.45, 0.97)	0.035	0.75 (0.52, 1.08)	0.118	0.74 (0.49, 1.14)	0.169
	266/21,257	No	1.00		1.00		1.00	
Blackberries	63/7631	Yes	0.58 (0.37, 0.91)	0.018	0.62 (0.40,0.97)	0.034	0.63 (0.40, 0.97)	0.036
	162/21,361	No	1.00		1.00		1.00	

Model 1: adjusted for age, sex, and race/ethnicity. Model 2: further adjusted for education level, poverty-to-income ratio, physical activity, current smoking status, depression severity, BMI, alcohol consumption, total energy, and caffeine intake. Model 3: further adjusted for education level, poverty-to-income ratio, physical activity, current smoking status, depression severity, BMI, alcohol consumption, total energy, caffeine intake, and modified HEI-2015.

## Data Availability

Data are available on the NHANES website, and the analytic codes will be made available pending email request to the corresponding author.

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
