# Peer review of "Berry Consumption and Sleep in the Adult US General Population: Results from the National Health and Nutrition Examination Survey 2005–2018"

_nutrients, 2023, doi:10.3390/nu15245115_

Round 1

Reviewer 1 Report

Comments and Suggestions for Authors

This is a very well done and written manuscript that expands a new area of emerging research.  I have only minor comments.

1. Please double check to ensure the manuscript is reported per the STRONE-nut guidelines.

2. It would be nice to have an idea of how many individuals (N(%)) reported taking melatonin supplements, since they are quite common.  This could be included in the participant characteristics.  All the authors decide on whether this could be used as a potential covariate.

3. The self-reported diabetes and hypertension variables are grossly accurate. There is a way to better calculate these variables that likely goes far beyond what is needed for this manuscript. I would suggest removing these two variables from the analyses and participant characteristics.

Author Response

Response to Reviewer Comments

1. Summary

2. Questions for General Evaluation

Reviewer’s Evaluation

Response and Revisions

Does the introduction provide sufficient background and include all relevant references?

Yes

Thanks!

Are all the cited references relevant to the research?

Yes

Thanks!

Is the research design appropriate?

Yes

Thanks!

Are the methods adequately described?

Yes

Thanks!

Are the results clearly presented?

Yes

Thanks!

Are the conclusions supported by the results?

Yes

Thanks!

3. Point-by-point response to Comments and Suggestions for Authors

Comments 1: Please double check to ensure the manuscript is reported per the STROBE-nut guidelines.

Response 1: Thank you for pointing this out. We agree with this comment. Therefore, we have examined our paper based on STROBE-Nutrition guidelines and we believe our paper follows the guidelines. We added this sentence to the methods and the corresponding reference: "We used the STROBE-nut checklist when writing our report [citation Lachat C, Hawwash D, Ocké MC, Berg C, Forsum E, Hörnell A, Larsson C, Sonestedt E, Wirfält E, Åkesson A, Kolsteren P, Byrnes G, De Keyzer W, Van Camp J, Cade JE, Slimani N, Cevallos M, Egger M, Huybrechts I. Strengthening the Reporting of Observational Studies in Epidemiology-Nutritional Epidemiology (STROBE-nut): An Extension of the STROBE Statement].

Comments 2: It would be nice to have an idea of how many individuals (N(%)) reported taking melatonin supplements, since they are quite common.  This could be included in the participant characteristics. All the authors decide on whether this could be used as a potential covariate.

Response 2: Thank you for your suggestions. The number and % of participants taking melatonin supplements are 369 (1.3%) in our sample. However, we decide not to consider melatonin use as a potential covariate and will not add it in the participant characteristics description because the addition of melatonin use as a covariate did not change any of the berry relationships with sleep outcomes. We have mentioned this in our originally submitted paper. Please refer to line: 274-275 in the revised paper.

Comments 3: The self-reported diabetes and hypertension variables are grossly accurate. There is a way to better calculate these variables that likely goes far beyond what is needed for this manuscript. I would suggest removing these two variables from the analyses and participant characteristics.

Response 3:  Agree. We have, accordingly, removed the variables: the history of hypertension and the history of diabetes in the participant characteristics description tables 1-3 and analyses related to Table 4-5, and modified the Tables 4-5 to emphasize this point. Please refer to the corrections/revisions highlighted/ in track changes.

4. Response to Comments on the Quality of English Language

Point 1: Language level acceptable.

Response 1:  Agree.

5. Additional clarifications

Response 1:  None.

Reviewer 2 Report

Comments and Suggestions for Authors

Based on the results of the US National Health and Nutrition Examination Survey from 2005 to 2018, the authors analyzed the relationship between berry consumption and sleep in the US adult population. However, it is necessary to sum up how many berries are appropriate to ingest, and how deep is more conducive to the sleep of the population? It is recommended that you modify it carefully. Other suggestions are as follows.

1. Twenty two percent is different from other direct numbers, please keep the full text consistent? Like 27%, 46%

2. Berries have other beneficial components due to the symbol polyphenols and polysaccharides, which are recommended not to be covered in the preface, but to refer to a published article, e.g. DOI: 10.1039/c9fo02171j,10.1007/s00394-022-02927-7

3. 3.3. Berry consumption associations with short or long sleep duration ;Is there a syntax error?

4. This data is 5 years old, can we add the latest data? (2019-2023?)

5. The authors need to further analyze the mechanism by which berries improve sleep.

6. The authors need to deeply analyze the relationship between berry intake and sleep, and recommend the amount of berries consumed by the human body.

 7. Does the interpretation of the results require a significance level? Please check whether the full text is marked?

Comments on the Quality of English Language

Language level acceptable.

Author Response

Response to Reviewer Comments

1. Summary

2. Questions for General Evaluation

Reviewer’s Evaluation

Response and Revisions

Does the introduction provide sufficient background and include all relevant references?

Can be improved

Revised.

Are all the cited references relevant to the research?

Can be improved

Revised.

Is the research design appropriate?

Can be improved

Revised.

Are the methods adequately described?

Can be improved

Revised.

Are the results clearly presented?

Can be improved

Revised.

Are the conclusions supported by the results?

Can be improved

Revised.

3. Point-by-point response to Comments and Suggestions for Authors

Comments 1: Twenty two percent is different from other direct numbers, please keep the full text consistent? Like 27%, 46%

Response 1: Thank you for pointing this out. We agree with this comment. Therefore, we have modified 22% to 22% of the participants in the highlighted/track changes of the resubmitted paper (line 186).

Comments 2: Berries have other beneficial components due to the symbol polyphenols and polysaccharides, which are recommended not to be covered in the preface, but to refer to a published article, e.g. DOI: 10.1039/c9fo02171j,10.1007/s00394-022-02927-7

Response 2: We have, accordingly, done the literature review of the recommended papers by the reviewer and have added a sentence about polysaccharides (See lines 56-57 in the resubmitted paper). As the types of berries consumed in the American diet are different than in these articles, we added a different reference on polysaccharides that we think is applicable to the American diet [Citation: Kelly Ross, Yaw Siow, Dan Brown, Cara Isaak, Lana Fukumoto & David Godfrey (2015) Characterization of Water Extractable Crude Polysaccharides from Cherry, Raspberry, and Ginseng Berry Fruits: Chemical Composition and Bioactivity, International Journal of Food Properties, 18:3, 670-689, DOI: 10.1080/10942912.2013.837066].

Comments 3: 3.3. Berry consumption associations with short or long sleep duration ;Is there a syntax error?

Response 3: Agree. We have, accordingly, modified the title 3.3 Berry consumption associations with short but not with long sleep duration, which was highlighted in the track changes in the resubmitted paper to emphasize this point.

Comments 4: This data is 5 years old, can we add the latest data? (2019-2023?)

 Response 4: We thank the reviewer for the suggestion; however, sleep and nutrition variables were limited or unavailable in the NHANES for these years proposed by the reviewer.

Comments 5: The authors need to further analyze the mechanism by which berries improve sleep.

Response 5: We thank the reviewer for the comments. We did incorporate melatonin use as a covariate; however, it did not change any of the berry relationships with sleep outcomes (refer to lines 276-278 highlighted in the resubmitted paper). We also acknowledge that the mechanisms by which berry consumption may affect sleep need further study in a future clinical trial which we addressed/highlighted in the discussion (see lines 376-378).

Comments 6: The authors need to deeply analyze the relationship between berry intake and sleep, and recommend the amount of berries consumed by the human body.

 Response 6: We appreciate the reviewer’s comments. The US Dietary Guidelines for Americans recommend about 2 cups of fruits per day for human health. Berries, as part of the recommended fruit servings, would likely help with improved sleep as the NHANES data in our study show that any berry consumption vs no consumption is beneficial. However, the exact quantity of berries that may benefit sleep would need to be determined in clinical trials. We added a statement to this effect in the discussion (Lines 376-378 in the resubmitted paper).

Comments 7: Does the interpretation of the results require a significance level? Please check whether the full text is marked?

 Response 7: Agree. We have, accordingly, removed the word significant in the discussion section (line 297 and line 350) to emphasize this point.

Comments/Suggestions 8 (in the summary comments): Based on the results of the US National Health and Nutrition Examination Survey from 2005 to 2018, the authors analyzed the relationship between berry consumption and sleep in the US adult population. However, it is necessary to sum up how many berries are appropriate to ingest, and how deep is more conducive to the sleep of the population? It is recommended that you modify it carefully.

Response 8: We appreciate the reviewer’s comments. The US Dietary Guidelines for Americans recommend about 2 cups of fruits per day. Berries could be included as part of these servings not only for sleep health but also for their other health benefits. The exact quantity of berries that may benefit sleep would need to be determined in clinical trials, but our studies indicate that any level of berry intake has a benefit over no berry consumption. As per above, we added a sentence in the discussion to address this issue (Lines 376-378 in the resubmitted paper).

4. Response to Comments on the Quality of English Language

Point 1: Language level acceptable.

Response 1:  Agree.

5. Additional clarifications

Response 1: None.
